# Amyloidosis Cutis Dyschromica in a 16-Year-Old Filipino Girl: A Case Report

**Fendi EJ R. Bautista \*, Marcia Marie S. Marte-Jimenez and Maria Jasmin J. Jamora**

Skin and Cancer Foundation Incorporated, Pasig City 1605, Philippines
* Correspondence: fendiejbautista@gmail.com

**Abstract:** Amyloidosis cutis dyschromica is a rare variant of primary cutaneous amyloidosis characterized by hyper- and hypopigmented macules. In this paper, we reported a case of a 16-year-old Filipino girl with hyper- and hypopigmented to depigmented macules on the upper and lower extremities, which started when she was 9 years of age.

**Keywords:** amyloidosis cutis dyschromica; amyloidosis; Congo red; hypopigmentation; hyper-pigmentation

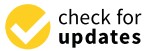



## 1. Introduction

Amyloidosis cutis dyschromica (ACD) is a rare variant of primary cutaneous amyloidosis (PCA) which was introduced by Morishima et al. in 1970. ACD is characterized by the presence of hyperpigmented, hypopigmented or depigmented macules of varying sizes, which can present in generalized fashion [1,2]. It is associated with no or little pruritus, prepubertal onset, and focal amyloid deposition in the papillary dermis [3].

## 2. Case Report

A 16-year-old Filipino girl was born to parents with a non-consanguineous marriage. At 9 years of age, hyperpigmented and hypopigmented macules were initially noted to appear on her legs. There was no associated redness, itch, pain or numbness, and no medication or product taken or applied. No consultation was conducted then. Her birth history was unremarkable. During the interim of 5 years, the patient noted a gradual increase in the number of the lesions which later on involved her arms. There was no associated pruritus, erythema, telangiectasia, atrophy or blisters. No prior episodes of photosensitivity, inflammatory cutaneous conditions or systemic illness were also reported. Her birth history and mental developmental milestones were normal. There was a questionable family history of hypopigmented macules on the legs of her maternal grandfather. The general physical examination was unremarkable. Laboratory tests requested were not performed due to financial constraints of the patient. No medication or product was applied to the affected areas prior to consultation. The patient had minimal sun exposure as she stayed mostly indoors. Cutaneous examination revealed bilateral symmetrical involvement of the distal aspect of upper and lower extremities [Figures 1 and 2] with lesions consisting of multiple, well-defined, dark brown to hypopigmented to depigmented macules on a background of hyperpigmented patches. Hair, teeth, nails, and oral mucosa were normal.

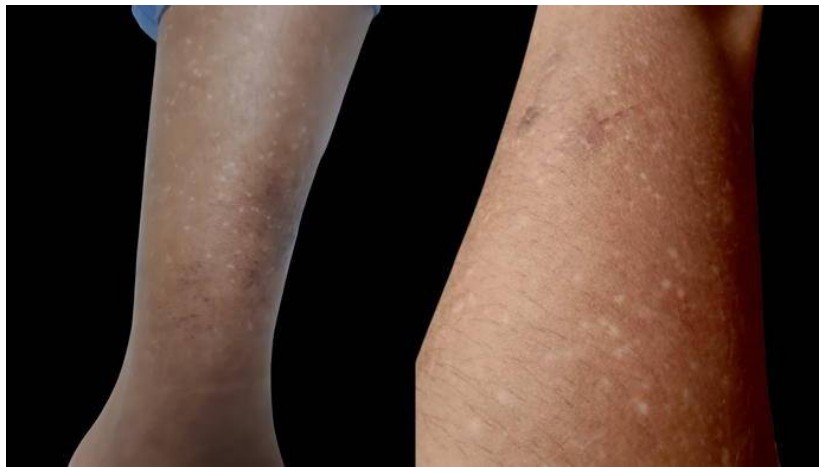

**Figure 1.** Hyperpigmented and hypopigmented to depigmented macules on the legs.

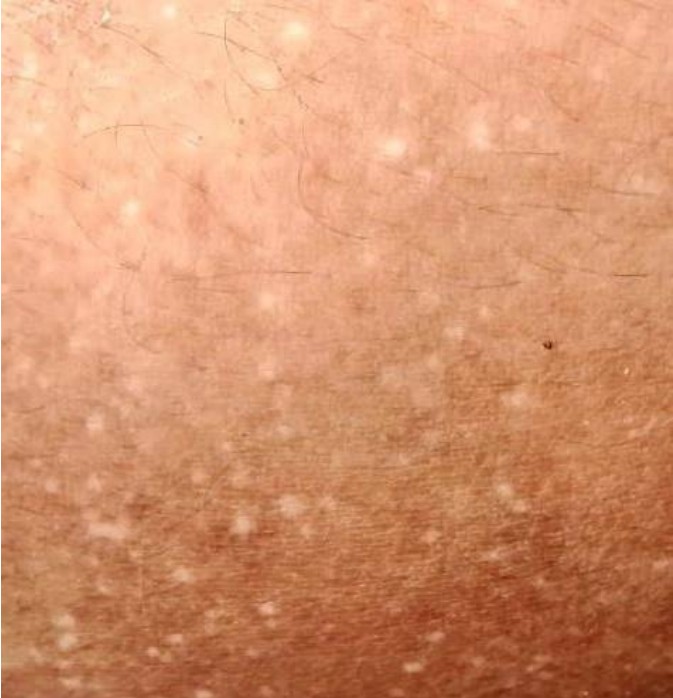

**Figure 2.** Close up photo of hypopigmented and hyperpigmented macules on the distal aspect of left arm.

A skin punch biopsy sample was taken from the hypopigmented macular lesion of her right leg. Histopathological examination stained with hematoxylin–eosin (H & E) stain, and observed under light microscopy, showed compact hyperkeratosis, hypergranulosis, and irregular psoriasiform hyperplasia. Within a widened papillary dermis, there are collections of amorphous pink globules with a perivascular infiltrate of lymphocytes, histiocytes, eosinophils, and melanophages [Figure 3A]. The presence of these amyloid deposits was confirmed by Congo red staining [Figure 3B]. Based on these typical clinical features, histopathological findings, and positive Congo red staining, the diagnosis of ACD was confirmed. After the diagnosis, the patient was advised to take vitamin C, avoid sun exposure, and apply sunscreen.

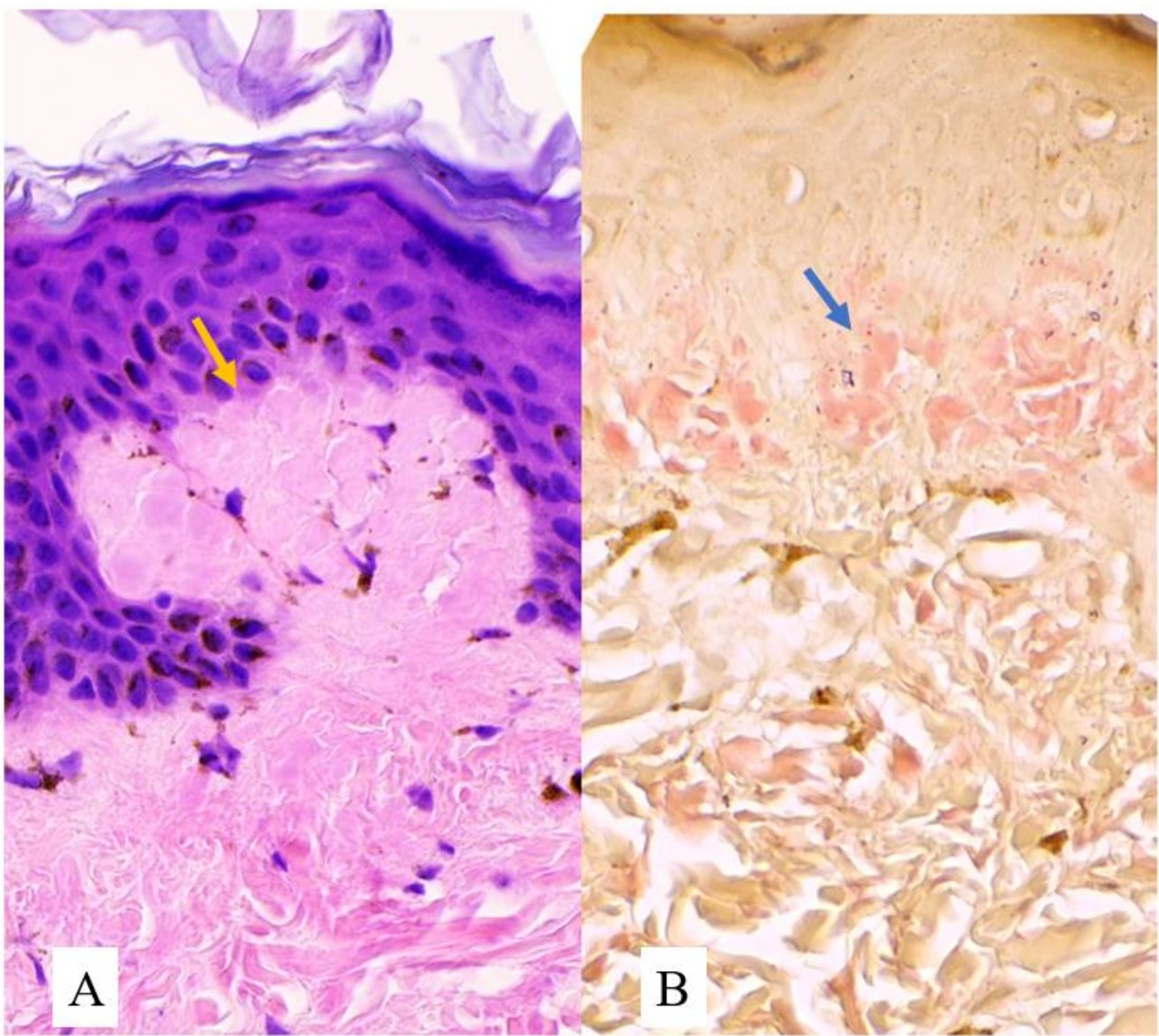

**Figure 3.** (**A**) Histopathology showed collections of amorphous pink globules (yellow arrow) and (**B**) Congo red staining revealed red congophilic deposits of amyloid in the papillary dermis (blue arrow) both under high power objective lens (400×).

### 3. Discussion

There are three major types of primary cutaneous amyloidosis, namely lichenoid, macular, and nodular. Amyloidosis cutis dyschromica belongs to atypical and rare variant of primary cutaneous amyloidosis, which was first discussed by Morishima et al. in 1970 [4].

It is predominantly reported in those of East and South-East Asian ethnicity (63%) [5]. Its main features include the following: (1) mottled hyperpigmented and hypopigmented macules, (2) beginning before puberty, (3) oftentimes absence of pruritus, and (4) amyloid deposits in papillary dermis histologically [4].

Reticular and dotted pigmentation with hypopigmented spots can spread all over the body, with or without associated pruritus [6]. Pathogenesis of amyloidosis cutis dyschromica is not clearly understood; however, it is assumed to be a familial disease with sunlight exposure as a major etiological factor. Ultraviolet B (UVB) and ultraviolet C radiations (UVC) lead to defective DNA repair and apoptosis of keratinocytes for individuals who are genetically predisposed. Histiocytes or fibroblasts phagocytose the apoptotic keratinocytes. As a result, cytokeratins of the lysed keratinocytes give rise to the amyloid material, and are then deposited in the skin [3,4,7,8]. In these ACD cases, Yang

and McGrath's groups have consistently reported glycoprotein non-metastatic melanoma protein B (GPNMB) gene mutations [9,10].

Several important entities may manifest in generalized pigmentation with hypopigmented macules. Differentials such as dyschromica, including dyschromatosis universalis hereditaria, xeroderma pigmentosum, tuberous sclerosis, idiopathic guttate hypomelanosis and poikiloderma-like amyloidosis are important to be considered. In histopathology of the former four conditions, amyloid deposits are not appreciated. Meanwhile, the latter shares similar ACD but other features such as poikiloderma, clinical features with lichenoid papules, bullae, photosensitivity, short stature, and palmo-plantar keratoderma are also present [8].

Avoidance of sun exposure and use of sun protection (e.g., sunscreen) are being advised to all the patients. Other treatment options, such as the use of topical corticosteroids, keratolytics, dimethyl sulfoxide, oral vitamin C and E and $CO_2$ laser have been studied, and showed varying results. For certain cases of ACD, systemic retinoic acid derivatives, particularly acitretin, have shown good improvement [4,11].

**Author Contributions:** Conceptualization, F.E.R.B., M.M.S.M.-J. and M.J.J.J.; validation, F.E.R.B., M.M.S.M.-J. and M.J.J.J.; investigation, F.E.R.B., M.M.S.M.-J. and M.J.J.J.; resources, F.E.R.B., M.M.S.M.-J. and M.J.J.J.; data curation, F.E.R.B., M.M.S.M.-J. and M.J.J.J.; writing- review and editing, F.E.R.B., M.M.S.M.-J. and M.J.J.J.; visualization, F.E.R.B., M.M.S.M.-J. and M.J.J.J.; supervision, M.M.S.M.-J. and M.J.J.J.; project administration, F.E.R.B., M.M.S.M.-J. and M.J.J.J. All authors have read and agreed to the published version of the manuscript.

**Funding:** This research received no external funding.

**Institutional Review Board Statement:** Ethical review and approval were waived for this study due to it not being required for case reports deemed not to constitute research.

**Informed Consent Statement:** Informed consent was obtained from all subjects involved in the study.

**Data Availability Statement:** No new data were created or analyzed in this study. Data sharing is not applicable to this article.

**Conflicts of Interest:** The Skin and Cancer Foundation, Inc. (SCFI) is a residency training institution accredited by the Philippine Dermatological Society (PDS), the only specialty society for Dermatology accredited by the Philippine Medical Association (PMA) and the Philippine College of Physicians (PCP). SCFI trains their graduates to be well-balanced and ethical dermatologists, as well as excellent clinicians, academicians and researchers. The following authors completed their residency training in dermatology in SCFI: Fendi EJ Robledo Bautista, Marcia Marie Solas Marte-Jimenez, Maria Jasmin Jacinto Jamora. In addition, Jamora is currently the training officer and section head of immunodermatology and dermatopathology of SCFI. Otherwise, all authors certify that they have no conflict of interest nor have they received any research grants.

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
