# Peer review of "Amyloidosis Cutis Dyschromica in a 16-Year-Old Filipino Girl: A Case Report"

_dermatopathology, doi:10.3390/dermatopathology10010002_

Round 1

Reviewer 1 Report

This is a well written case report of amyloidosis cutis dyschromica (ACD) in an Asian female. This is a relatively rare entity, particularly in regards to incidence in Asian population.  

For better clarification, it would be helpful to have arrows indicating the key histologic features in Figures 3 and 4.  Moreover, low and high magnifications of the histology images should be included, including stating the magnification (such as X20 or X200).  More developed discussion of ACD in the literature and findings (clinical and histologic) would help strengthen the article.

Author Response

Thank you very much for your comments and suggestions. Kindly see below for the point-by-point response:

Figures 3 and 4 have been reoriented and merged for better viewing. Arrows indicating the key histologic features were also added.  Moreover, low and high magnifications of the histology images stating the magnification has been included. Discussion was also improved by adding clinical differential diagnosis, as well as skin biopsy findings and key pathologic features.

Thank you very much! 

Reviewer 2 Report

This is a typical case of amyloidosis cutis dyschromica (ACD). It was well-written and the image quality is good. There are a few comments.

1. The clinical images are clear but image 1 and 2 can be merged into one. Please reoriente the pathology images (Fig 3 and 4) and make the epidermis parallel to the top, not oblique. This two images can also be merged into one picture.

2. ACD is a cutaneous amyloidosis. It's better to provide a cytokeratin stain to demonstrate the amyloids are positive for cytokeratin and exclude light chain or other systemic amyloidosis.

3. There are several important entities which may manifests in generalized pigmentation with hypopigmented macules, for example, dyschromatosis universalis hereditaria and generalized Dowling-Degos disease. It's better to add clinical differential diagnosis in the discussion and emphasize the importance of skin biopsy and pathological diagnosis. 

4. The genetic alternation in ACD was firstly reported in the American Journal of Human Genetics (102, 219–232, 2018) "Loss of GPNMB Causes Autosomal-Recessive Amyloidosis Cutis Dyschromica in Humans". It's better to cite this article in references.

5. The typesetting in the discussion needs to be improved.

Author Response

Thank you very much for your comments and suggestions. Kindly see below for the point-by-point response:

1. Figures 3 and 4 have been reoriented and merged.

2. The authors agree that it is better to provide a cytokeratin stain; however, due to limited means, we were not able to add cytokeratin stain findings for this case in the revised manuscript. 

3.  Clinical differential diagnosis, as well as skin biopsy findings and key pathologic features, have been added in the discussion.

4. "Loss of GPNMB Causes Autosomal-Recessive Amyloidosis Cutis Dyschromica in Humans" by Yang et. al. has been cited and added in the references.

5. The typesetting in the discussion has been revised.

Thank you very much! 

Round 2

Reviewer 2 Report

No further comments.

Author Response

Thank you very much for your comments and suggestions.